# Improving the Quality of Frozen Lamb by Microencapsulated Apple Polyphenols: Effects on Cathepsin Activity, Texture, and Protein Oxidation Stability

**DOI:** 10.3390/foods11040537

**Published:** 2022-02-14

**Authors:** Yuanyuan Zhong, Yangming Liu, Lijie Xing, Mou Zhao, Wenxia Wu, Qingling Wang, Hua Ji, Juan Dong

**Affiliations:** 1College of Food Quality and Safety, Shihezi University, Shihezi 832000, China; zyy18290775548@163.com (Y.Z.); zm13026091096@163.com (M.Z.); wwx2512467283@163.com (W.W.); qingling1100@163.com (Q.W.); jh_food@shzu.edu.cn (H.J.); 2Sesame Research Center, Henan Academy of Agricultural Sciences, Zhengzhou 450002, China; pl20200042@163.com; 3Analysis and Testing Center, Xinjiang Academy of Agriculture and Reclamation Science, Shihezi 832000, China; xiaolongnv2088@163.com

**Keywords:** freezing process, microcapsules, polyphenols, oxidation stability, meat system

## Abstract

This study aimed to evaluate the efficiency of microencapsulated apple polyphenols (MAP) in controlling cathepsin activity and texture, as well as inhibiting protein oxidation and metmyoglobin formation in lamb meat during frozen storage at −18 °C for 40 weeks. The effects of degradation in vitro on cathepsin and the microstructure in lamb were also evaluated. Results indicated that relative to the control group, the lamb treated with MAP exhibited increased cathepsin activity and inhibited metmyoglobin production. Textural characteristics, such as hardness and springiness, significantly changed (*p* < 0.05). Treatment with 0.2–1.6 mg/mL of MAP effectively reduced the mean particle size, increasing the zeta potential, delaying the conversion of α-helices to random coils, and maintaining the integrity of the tissue structure. However, treatment with 3.2 mg/mL of MAP damaged the protein structure. Degradation in vitro indicated that protein oxidation hindered the effect of cathepsin and was a dominant factor affecting protein during the frozen storage. These results demonstrated that microencapsulation can potentially be used for meat preservation and replace chemical antioxidants in the meat industry.

## 1. Introduction

Lamb is one of the most popular types of meat consumed in China and other countries worldwide. It is a valuable source of high-quality protein and is preserved throughout export mainly in frozen form and for market trade circulation [1,2]. Biochemical reactions are significantly slowed down during the frozen storage; nonetheless, oxidation still occurs under freezing conditions. Consequently, the texture deteriorates, limiting the marketability of lamb meat stored for an extended period [3]. According to previous studies, meat deterioration with the extension of freezing time could be mainly attributed to three major processes: (i) An oxidative process occurs, which allows oxygen-free radicals to accumulate, initiating damage to protein and lipid oxidation. Protein oxidation is also suggested to induce protein cross-linking and aggregation, which directly affect the functional properties of protein [4,5]; (ii) Endogenous proteolytic activities change, which leads to protein degradation during meat storage, inducing the rupture of the muscle cytoskeleton [6]; (iii) The formation and growth of ice crystals during cryopreservation destroy the muscle fiber structure and cells, which can aggravate the mechanical damage to muscle tissue [7]. These factors lead to quality deterioration, as indicated by discoloration, off-flavors, texture deterioration, and a reduction in the nutritional value, resulting in the loss of consumer acceptability [8].

The search for natural antioxidants has recently drawn considerable research interest, driving the meat industry to increase the shelf life of their products. With potent antioxidant activities, polyphenols can delay lipid and protein oxidation in meat and meat products. They play a specific role—for instance, breaking the oxidative chain reaction and scavenging free radicals [9]. However, polyphenols are extremely unstable under inappropriate heat, moisture, and light conditions because of the existence of active phenolic hydroxyl in their molecular structure. Several studies have investigated the covalent and noncovalent interactions between phenolics and meat protein, which can modify the structure, conformation, and physicochemical properties of protein, thereby changing the function of protein [10,11]. Currently, microencapsulation is a technology that can effectively protect the stability and activity of polyphenols [12,13]. Studies indicated that the oxidation and color stability of dried minced pork slices were markedly enhanced when mulberry polyphenol microcapsules were used as additives [14]. Microencapsulated pitaya peel extracts were introduced by Cunha et al. [15] to mitigate the adverse effects of UV-C radiation on the oxidative stability of ground pork patties. The stability of bayberry polyphenols was shown to be improved by microencapsulation [16]. Several microencapsulated plant polyphenols have been investigated, but no studies have thus far been reported on the microencapsulation of apple polyphenols (MAP); moreover, the potential of microencapsulation for improving the quality of meat during frozen storage is yet undetermined.

In the current study, apple polyphenols (APs) were microencapsulated by hydroxypropyl-β-cyclodextrin, and the effect of MAP on the quality of frozen lamb during storage was evaluated. Changes in cathepsin activity, texture characteristics, and protein oxidation stability were compared to analyze the protective effect of MAP at different concentrations on lamb meat. In addition, in vitro degradation of cathepsin and microstructure in lamb were investigated to elucidate the results of improving the quality of frozen meat.

## 2. Materials and Methods

### 2.1. Materials

Ten fresh hindquarters from different randomly selected lambs (7–8 months old) were purchased from Xinjiang Western Animal Husbandry Co., Ltd. (Shihezi, China). The breed was Kazakh lamb fed on Tianshan pasture, which was mainly fed with alfalfa in free pasture. Apple polyphenols (APs, purity > 75%, containing phloridzin, ellagic acid, procyanidin B2, epigallocatechin, etc.) were supplied by Yuanye Bio-Technology (Shanghai, China). 2-Hydroxypropyl-β-cyclodextrin, E-64-protease inhibitor, cathepsin B (cath-B) (Product number: C8571), and cathepsin L (cath-L) (Product number: C6854) from the human liver, and tris were purchased from Sigma-Aldrich (Shanghai, China). Sheep cath-L and cathepsin B (cath-B) ELISA Kit were supplied by Jiangsu Jingmei Biotechnology Co., Ltd (Suzhou, China).

### 2.2. Preparation of Frozen Samples

One fresh gigot (hindquarter primal, *n* = 10; weight, 2 ± 0.2 kg) was sampled per lamb (*n* = 10). The samples were purchased within 24 h from ripening treatment and then transported in an ice bath to the laboratory. The fat, fascia, and connective tissue of the samples were removed and cut into meat pieces of the same size and quality (length and width, 10 ± 1.5 cm; weight, 100 ± 2.5 g; *n* ≥ 8 per lamb). Three samples (*n* = 3) were separated for the initial analysis of fresh meat. Microcapsules were prepared as described in our previous studies, [17] by mixing APs and hydroxypropyl-β-cyclodextrin in at the mass concentration of 1:1 via freeze drying in the laboratory (the highest encapsulation efficiency of 82.31% and the highest polyphenol loading rate of 46.87% were reached under this condition), obtaining the solutions S1, S2, S3, and S4 with concentrations of 0.2, 0.8, 1.6, and 3.2 mg/mL, respectively. The remaining samples (*n* = 75) were randomly divided into 5 groups: Group 1 received no treatment (control), whereas Groups 2–5 were soaked in MAP solutions of S1, S2, S3, and S4 for 60 min (4 °C) and then drip dried in a stainless-steel orifice plate (5 min). Each batch contained at least 15 samples, covered with oxygen permeable polyvinylchloride (PVC) film (length, 15 cm; width, 17 cm) and frozen stored at −18 °C for 40 weeks. After thawing at 4 °C for 12 h, the samples were further analyzed.

### 2.3. Determination of Cathepsin Activity in Lamb during Frozen Storage

The activity of cathepsin (B, L) was determined as specified in the kit. First, the meat samples were treated with phosphate-buffered solution (pH 7.0), and corresponding reagents were added, following the procedure specified in the ELISA kit. The absorbance was measured at 450 nm, and the enzyme activity was calculated (U/mL).

### 2.4. Determination of Deoxymyoglobin, Oxymyoglobin, and Metmyoglobin Contents

DMb, OxyMb, and MetMb contents were measured as described by Carlez, Veciana-Nogues, and Cheftel [18]. Meat samples (2 g) were weighed and added with 20 mL of phosphate solution (0.04 mol/mL, pH 6.8), homogenized, and centrifuged for 30 min (4 °C). The supernatant was diluted and filtered to determine the absorbance levels at 525, 545, 565, and 572 nm. The contents of DMb, OxyMb, and MetMb were calculated using the following formula:(1)$$\mathrm{DMB}\;\left(\%\right)=\left({0.369R}_{1}+{1.14R}_{2}-{0.941R}_{3}+0.015\right)\times 100$$
(2)$$\mathrm{OxyMb}\;\left(\%\right)=(0.882R_{1}+1.267R_{2}+0.809R_{3}-0.361)\times 100$$
(3)$$\mathrm{MetMb}\;\left(\%\right)=(-2.541R_{1}+0.777R_{2}+0.8R_{3}+1.098)\times 100$$
(*R*_1_ = A_572nm_/A_525nm_, *R*_2_ = A_565nm_/A_525nm_, *R*_3_ = A_545nm_/A_525nm_)

### 2.5. Texture Analysis

The texture characteristics of lamb meat were measured using a texture analyzer (TA-XT Plus, Stable Micro Systems, Surrey, UK) as reported by Zang, Xu, Xia, and Jiang [19]. A P/5 (diameter 5 mm) flat-bottomed cylindrical probe was used to simulate human teeth chewing food. The samples were compressed twice and then tested in TPA mode. The following texture profile attributes were measured: (i) hardness (kg), which corresponds to the peak force required for the first compression cycle and (ii) springiness (mm), the distance the sample retrieves after the first compression. The experiment was repeated 8 times in each group, and the average was determined after the abnormal value was eliminated.

### 2.6. Extraction of Myofibrillar Proteins

The myofibrillar protein (MP) was extracted, following the method used by Lefevre et al. [20], with a slight modification. A standard phosphate-buffered solution (100 mmol/L NaCl, 1 mmol/L EDTA; pH 7.0) was added to lamb meat. The mixture was homogenized three times to remove the water-soluble protein and other substances and then washed with a phosphate-buffered solution (0.6 mol/L NaCl, pH 7.0) and placed in an ice bath for 2 h. The protein suspension was ultimately filtered through four layers of clean and dry gauzes to remove the connective tissue. The MP concentration was measured using the biuret technique, using the method reported by Gornall, Bardawill, and David [21].

### 2.7. Zeta Potential Measurement

The zeta potential of the MP samples was determined by combining the backscattering and dynamic scattering of the zeta potential analyzer (Nanoplus-3, Brookhaven Instruments, Holtsville, NY, USA). MP samples were dispersed using the phosphate buffer (25 mmol/L, pH 7.0) and measured at 25 °C. Distilled water was used as the solvent, and each sample was repeated three times.

### 2.8. Particle Size Measurement

The particle size distribution of the MP was analyzed using a laser particle size analyzer (S3500, Microtrac Inc., Montgomeryville, PA, USA), and the protein concentration was adjusted to 3 mg/mL. The specific parameters were as follows: refractive index of the material, 1.520; medium, reverse-osmosis water; refractive index, 1.333; absorption coefficient, 0.01; and measured particle sizes, the surface-weighted mean diameter after 5 cycles.

### 2.9. Raman Spectroscopy

A laser microscopic Raman imaging spectrometer (Bruker SENTERRA, Bruker Technology Co., Ltd., Billerica, MA, USA) equipped with a DXR 785 nm laser source was used to analyze the secondary structure of MP. The spectra were recorded between 600 and 3300 cm^−1^ under the following conditions: exposure time, 45 s; slit width of the diaphragm, 50 μm; resolution ratio, 6 cm^−1^; and number of sample exposures, 3. Protein secondary structures in these samples were determined as percentages of α-helix, β-sheet, random coil, and turn conformations by using the method described by Lee and Wallace [22].

### 2.10. In Vitro Degradation of Myofibrillar Protein by Cathepsin

The approach was designed based on previous studies [23], with certain modifications. The MP was diluted, reducing the concentration to 2 mg/mL with 50 mmol/L acetate buffer (containing 2 mmol/L EDTA, 0.1 mol/L KCl, pH = 5.5). Cath-B and cath-L were fully dissolved in 50 mmol/L citrate phosphate buffer (pH = 5.5) to 0.01 μg/μL and stored in the refrigerator (−20 °C). The MP (0.5 mL) and cathepsin solution (0.1 mL) were mixed and hydrolyzed in a water bath (37 °C) for 1 h. Finally, E64 (0.01 mol/L) was added to terminate the reaction. After centrifugation, the precipitate was fully dissolved in a Tris urea solution (40 mmol/L), mixed with a loading buffer solution, and heated. Sodium dodecyl sulfate-polyacrylamide gel electrophoresis (10% separation gel and 5% concentrated gel) was performed.

### 2.11. Hematoxylin-Eosin Staining

The frozen sample was cut into 5 mm × 5 mm × 5 mm pieces and fixed with paraformaldehyde (4%) at room temperature for 48 h. The fixed sample was immersed in a container filled with liquid wax and then sliced into sections 4–7 μm thick after the paraffin was solidified. The sliced samples were stained with hematoxylin for 5 min, washed gently with distilled water, differentiated for 60 s, soaked in distilled water for 10 min, stained in eosin for 2 min, and rinsed lightly with distilled water for 3 min. After dyeing, the slices were dehydrated with 100% ethanol and then rinsed with xylene, rendering them transparent. Finally, the sample was sealed with neutral glue and observed under a light microscope (CX41, Olympus Co., Ltd., Shenzhen, China).

### 2.12. Statistical Analysis

Each experiment was conducted in triplicate measurements for three replicates, with the result expressed as the mean value ± standard deviation. All data were conducted using one-way ANOVA, followed by the Duncan procedure between the means by using SPSS 21.0 (SPSS Inc., Chicago, IL, USA). Significance was defined at *p* < 0.05. The graphs were generated using Origin 8.5 (Northampton, MA, USA) and Adobe Illustrator CS5 (San Jose, CA, USA).

## 3. Results and Discussion

### 3.1. Effect of Different Treatments on Cathepsin Activities in the Frozen Lamb

As shown in Figure 1a, cath-B activity in each treatment group is increased at Week 2 relative to that at the beginning of the storage period (0.47 U/mL). This increase might be attributed to freezing and thawing, which caused ice crystals to puncture the lysosomal membrane. Consequently, cathepsin was released from the lysosome to the myofibril, and the effect was greater than the inhibition of low temperature on cathepsin activity. Moreover, enzyme activity in the MAP treatment group reached the peak at Week 8, compared with the control; the S3 group exhibited the highest activity (0.72 U/mL, *p* < 0.05). This occurrence, which may be related to the gradual diffusion of the preservation solution in meat, stimulated the lysosome and affected enzyme activity. Subsequently, cath-B activity decreased with an increase in frozen storage time. At Week 40, the corresponding decreases in cath-B activity were as follows: control group, 36.17%; S1, 31.91%; S2, 12.77%; S3, 4.25%; and S4, 8.51%. Notably, cath-B activity was significantly lower in the control group than in the MAP-treated groups (*p* < 0.05). Frozen storage may inhibit enzyme activity, or cathepsin in the lysosome is released into other subcellular structures [24].

A similar trend for cath-L activity is also depicted in Figure 1b. During frozen storage, cath-L activity in all groups significantly decreased to varying degrees (*p* < 0.05). These results were consistent with the study by Nagaraj and Santhanam [25] in which the cath-L activity decreased significantly when goat muscles were stored at −15 °C for 16 weeks. Changes in cell membrane permeability, structure, and function caused by free radical attack during long-term frozen storage have led to the denaturation or inhibition of cathepsin activity [26]. However, treatment with MAP could significantly delay the decline in the cath-L activity of lamb muscle, which remained higher in the MAP-treated group with S3 than in other groups (S1, S2, and S4) during frozen storage (*p* < 0.05). In addition, cath-L activity was less than cath-B activity during the freezing period, indicating that cath-B exhibited dominance over cath-L during frozen storage.

### 3.2. Effects of Different Treatments on the Contents of Deoxymyoglobin, Oxymyoglobin, and Metmyoglobin in the Frozen Lamb

To assess the effect of MAP treatment on the meat color of frozen lamb, changes in the contents of DMb, oxymyoglobin (OxyMb), and metmyoglobin (MetMb) were measured (Figure 2). The DMB and OxyMb contents of all treatments significantly decreased during frozen storage, where the MetMb content increased (*p* < 0.05). The changes occurred rapidly within the first 16 weeks of storage. The bright-red color of meat is generally attributed to the formation of OxyMb. The relative contents of DMB, OxyMb, and MetMb in meat mostly depend on the availability of oxygen, the automatic oxidation rate of myoglobin, and the reduction potential of MetMb [27]. During frozen storage, DMB was oxidized to OxyMb and then converted to MetMb, thereby reducing the stability of the meat color. The primary and secondary products of lipid oxidation may also promote the accumulation of MetMb. In the current study, the change in the myoglobin content in the MAP treatment was preferable to that in the control group, which indicated that polyphenols in MAP could maintain the stability of meat color by slowing down the loss of DMB and OxyMb contents and preventing the increase in MetMb; alternatively, phenolic substances could inhibit the oxidation of lipid and OxyMb in meat. In addition, the MetMb content was significantly lower in the MAP treatment group with S3 than in other MAP treatment groups (S1, S2, S4) during the frozen storage period (*p* < 0.05), indicating that the samples treated with S3 improved the stability of meat color. Other previous studies also showed that the addition of a pistachio green shell extract could effectively reduce the formation rate of MetMb content in beef patties during chilled storage [28]. Ganhão, Morcuende, and Estévez [29] similarly reported color stability of the Rosa canina extract in burger patties during refrigerated storage.

### 3.3. Effect of Different Treatments on the Texture of Lamb Meat during Frozen Storage

The decrease in hardness or other related texture parameters, such as springiness, in frozen control lamb could largely affect consumer acceptability and is thus highly undesirable. The effects of different treatments on the texture characteristics of frozen lamb are listed in Table 1. With the prolongation of frozen storage time, muscle hardness in each treatment group decreased significantly (*p* < 0.05). This reduction may be due to the continuous growth of ice crystals during frozen storage, squeezing the muscle cells and causing mechanical damage to the cells. This process destroys the integrity of the tissue structure and water loss, resulting in a decrease in muscle hardness. Wang et al. [30] also found that endogenous proteases (such as calpain and cathepsin) can act on myofibrillar protein, promote its degradation, and affect muscle hardness. After Week 40 of frozen storage, the hardness values of the control, S1, S2, S3, and S4 groups were only 36.9%, 49.8%, 48.8%, 63.1%, and 55.9% of the initial values (0.84 kg), respectively. Springiness reflects the degree of deformation of food under external force and the degree of recovery after withdrawal. As presented in Table 1, the springiness of fresh lamb (frozen for 0 d) is 1.03 mm, showing a significant downward trend during frozen storage (*p* < 0.05); meanwhile, the springiness of the lamb in the MAP-treated groups was significantly higher than that in the control group (*p* < 0.05). This difference may be attributed to the decrease in muscle water content and protein decomposition, softening the muscle and reducing springiness. This effect is consistent with that reported by Sun, Zhang, Bhandari, and Yang [31] and Wei et al. [32], who observed reductions in the springiness of crayfish and tilapia fillets during frozen storage for 42 and 49 d, respectively. However, the texture parameters of frozen lamb decreased slowly in different MAP treatment groups. Such decreases may be attributed to the physicochemical modification of polyphenols during frozen storage, which reduces the force between cells and inhibits muscle decomposition. Thus, MAP treatment can delay texture deterioration in frozen lamb. Among the treatments, the S3 treatment exerted the strongest effect on improving hardness and springiness.

### 3.4. Effects of Different Treatments on Protein Stability during Frozen Storage

#### 3.4.1. Zeta Potential

The absolute value of zeta potential is closely related to the stability of protein aggregation, and a high absolute value indicates that the protein has enhanced stability, which can regulate the interaction between protein molecules [33]. Figure 3A(ii) shows that the absolute value of zeta potential is significantly decreased in the control group at Week 40 relative to that in the fresh sample (*p* < 0.05), indicating a decrease in the electrostatic interaction with an increase in freezing time. The absolute value of the zeta potential of the fresh sample was 25.62 mV, which was 2.95 times that of the control group. Arredondo-Parada et al. [34] suggested that the change in potential difference could be related to the conformational change in some proteins, leading to the exposure of hydrophobic nonpolar residues. The addition of different MAP treatment groups relatively enhanced the electrostatic repulsion, reduced the degree of coagulation between MP solutions, further prevented the formation of protein aggregates, and improved the stability of protein. Notably, the potential distribution of the control group was significantly widened (Figure 3A(i)). The non-uniformity of charge distribution generally leads to the imbalance of partial charge between protein molecules, which reduces the stability of MP.

#### 3.4.2. Particle Size

Particle size can directly reflect the size distribution of soluble protein and the aggregation of myofibrillar protein oxidation [35]. As shown in Figure 3B(i), the distribution range of the particle size increases after frozen storage for 40 weeks in each treatment group relative to that of the fresh samples and tends to move toward the direction of increasing the particle size. This observation may be attributed to the formation of large polymers via disulfide bond formation between actomyosin molecules. However, the size distribution of the protein solution with different MAP treatments was more concentrated and narrower during storage for 40 weeks. Consequently, the particle size was more uniform, indicating that the addition of MAP could delay the aggregation of myofibrillar protein during frozen storage. This trend of treatment effects was consistent with that of the zeta potential. In addition, the mean particle diameter of the control group (Figure 3B(ii)) increased by 68.2% relative to that of the fresh samples, indicating that protein oxidation can promote the formation of insoluble components; meanwhile, the mean particle diameter was significantly reduced in the MAP treatment group (*p* < 0.05). These results demonstrate that the conversion of MAP treatment can reduce the degree of protein oxidation and protect the particle size distribution of small molecular proteins. Similar results were found in a study in which adding sodium pyrophosphate reduced the increase in particle size caused by protein oxidation [36].

#### 3.4.3. Secondary Structure in Myofibrillar Proteins

The conformational information of the backbone and side chain can be obtained by Raman spectroscopy for the chemical groups of proteins, such as the amide bond and C-C skeleton stretching vibration. Therefore, the changes in the amide I band in Raman spectra were detected to characterize the secondary structure of proteins. Different peaks assigned to different secondary structures were as follows: the amide I band focused on 1650–1660 cm^−1^ for α-helix, 1665–1680 cm^−1^ for β-sheet, 1660–1665 cm^−1^ for random coil and β-turn [37,38]. As shown in Figure 4a, the peak intensity of the control group is significantly higher than that in the fresh group after frozen storage for 40 weeks. The reason may be that the protein was attacked by hydroxyl radicals, which changed the secondary structure of the MP. This change was also inhibited to a certain extent by adding different MAP treatment groups, which may be attributed to the ability of polyphenols to bind to protein via covalent cross-linking, blocking the action site of reactive oxygen species [39]. The percentages of the MP secondary structures corresponding to different treatments are presented in Figure 4b. The results showed that the α-helix content of the fresh samples was 40.1%, which sharply reduced the percentages of the control group to 19.3% after frozen storage for 40 weeks; meanwhile, the ratios of β-turns and random coils increased markedly to 22.1% and 43.6%, respectively. The decrease in the α-helix of the MP may be related to the hydrogen bonds breaking gradually and the partial unfolding of the helical structure under low temperatures [40]. Concomitantly, the ratio of α-helix in the samples treated with MAP decreased by 0.6–20.5%, which slowly decreased at the end of the storage relative to that in the control group. The content of α-helix treated with S3 exhibited the highest stability. On the one hand, microencapsulation of polyphenols may weaken the breaking of hydrogen bond, delay the formation of the intermolecular disulfide bond, and enhance the stability of the MP secondary structure during frozen storage. On the other hand, driven by electrostatic hydrophobic interactions, phenolic compounds interact with protein, improving protein stability; moreover, they maintain a higher α-helix content and lower random coils [41]. However, the addition of S4 further destroyed the secondary structure of the protein, which is consistent with the observation reported by Jia, Wang, Shao, Liu, and Kong [42], who observed that high concentrations of catechin could reduce the structural stability of MP under oxidative stress.

### 3.5. SDS–PAGE Analysis of Myofibrillar Protein Degradation by Cathepsin In Vitro

Cathepsins (B, L) are used to hydrolyze MPs in vitro. The effects of protein oxidation corresponding to different treatments on cathepsins during frozen storage are presented in Figure 5. For cath-B enzymolysis (Figure 5A), the intensity of N-CK bands from 24 w to 40 w gradually decreased, indicating that the degree of molecular cross-linking or degradation caused by protein oxidation increased with the extension of the freezing storage time [43]. Clearly, after cath-B was added, the CK group showed that the myosin heavy chain (MHC), actin, tropomyosin (Tm), and myosin light chain (MLC) bands were hydrolyzed, and Tm and MLC bands after 24 weeks were almost completely degraded. However, the MAP-treated groups (S1–S3) were likely to maintain their structure during frozen storage for up to 40 w; thus, the protein bands representing them maintained their intensities. Moreover, the intensity of the MHC bands in the different treatment groups with cath-B at Week 24 was lower than that at Week 40, indicating that the sensitivity of MHC to cathepsin hydrolysis was generally reduced after frozen storage; meanwhile, the MAP-treated groups showed a gradual decrease in MHC and actin to varying degrees. Slight decreases in the intensity of protein bands in the S4-treated samples were determined relative to the S1–S3 samples. Similarly, the hydrolysis of cath-L is shown in Figure 5B. Therefore, the degree of hydrolysis in different groups after the addition of cathepsins at Week 24 was higher than that at Week 40, and MHC and actin were more apparent in the S4 group. These observations suggest that the structure of the protein changed during frozen storage, and the protein structure at Week 24 only exposed the action site of cathepsin and thus could play a better role in hydrolysis [23]. As previously mentioned, MP was found to be oxidized with the extension of frozen storage, thereby reducing the degree of hydrolysis by cathepsin and indicating that MP oxidation reduced the sensitivity to cathepsin. Therefore, these results demonstrate that the conversion of protein oxidation plays a major role in frozen storage, and cathepsin plays an auxiliary role.

### 3.6. Effects of Different Treatments on the Microstructure during Frozen Storage

The increase in the volume of ice crystals during frozen storage can cause the compression and deformation of muscle fibers. The loss of muscle tissue juice increases after thawing, ultimately affecting the sensory quality and nutritional value of meat. Changes in hematoxylin–eosin staining in the cross-section (A) and longitudinal section (B) of frozen lamb muscle tissue after different MAP treatments are presented in Figure 6. At the initial storage time (Day 0), the connective tissues in the fresh lamb were arranged uniformly and tightly, and the cell boundaries were quite distinct with small gaps. The tissue structure of the control group changed significantly after frozen storage for 40 weeks, compared with the initial sample. The muscle fiber structure changed from dense to loose; the gap between the inside and outside of the muscle bundle (white) increased [44]; and pores of different sizes appeared in the cross-section and longitudinal section. These alterations may be explained by the destruction of muscle tissue cells by ice crystals during cryopreservation, causing mechanical damage and increasing the spacing between or within muscle bundles [45]. Notably, MAP treatment of S1–S3 groups significantly inhibited structural changes. Specifically, in the S3-treated group, small and fewer gaps were maintained, and the integrity of the muscle tissue was improved, without apparent squeezing and distortion. An appropriate concentration of polyphenols, combined with microencapsulation treatment, evidently helped inhibit the damage to the frozen lamb muscle tissue. In other previous studies, tangerine peel extract coating significantly hindered the changes in the tilapia tissue structure and maintained small voids during frozen storage [46]. Zhang et al. [47] also observed that shrimp soaked in a sugar solution inhibited the growth of ice crystals and reduced mechanical damage to the structure during freezing, thereby stabilizing the integrity of the shrimp tissue structure. By contrast, the S4-treated group showed a looser tissue structure than those of the S1–S3 groups, which was consistent with the results of the aforementioned protein stability study. Therefore, the protective effects of microencapsulated polyphenols on the ability to maintain cathepsin activity and myofibrillar protein stability were observed during frozen storage, in addition to the retention of muscle tissue integrity. These changes resulted in reduced hardness and springiness, restraining the accumulation of metmyoglobin.

## 4. Conclusions

Microencapsulation treatment with APs and hydroxypropyl-β-cyclodextrin showed excellent performance in controlling the loss of quality and changes in cathepsin and protein in frozen lamb. Treatment with 0.2–1.6 mg/mL of MAP effectively maintained the uniformity of the charge distribution and particle size aggregation of the MPs, as well as delayed the oxidative denaturation of protein compared to the control. MAP also significantly affected the secondary structure of MPs by inhibiting protein–molecular interactions, resulting in enhanced α-helix stability. Meanwhile, MAP treatment maintained the cathepsin activity and delayed the accumulation of methemoglobin, thereby preventing meat discoloration and maintaining the stability of meat texture. Further, the structure of the muscle tissue of the lamb treated with 1.6 mg/mL MAP was relatively compact and complete, as shown by microstructural analysis. However, treatment of lamb with 3.2 mg/mL of MAP led to a decrease in the stability of the protein despite the reduction in methemoglobin; thus, reduced concentrations are recommended for meat preservation. In addition, in vitro degradation indicated that protein oxidation was the dominant factor driving meat quality loss during the frozen storage. Therefore, this study demonstrated the potential of microencapsulation for enhancing the potential of APs to improve the oxidative stability of protein and textural characteristics in frozen meat.

## Figures and Tables

**Figure 1 foods-11-00537-f001:**
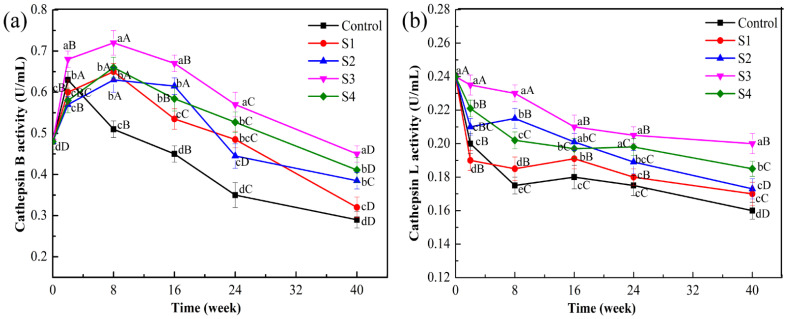
Activities of cathepsin B (**a**) and cathepsin L (**b**) from lamb meat with different treatments during frozen storage. (MAP: microencapsulated apple polyphenols; Control: group without treatment; S1–S4: samples treated with 0.2, 0.8, 1.6, and 3.2 mg/mL of MAP). Letters (A–D) represent a significant difference (*p* < 0.05) between the frozen storage periods in the same treatments. Letters (a–d) represent a significant difference (*p* < 0.05) between the different treatments in the same frozen storage periods.

**Figure 2 foods-11-00537-f002:**
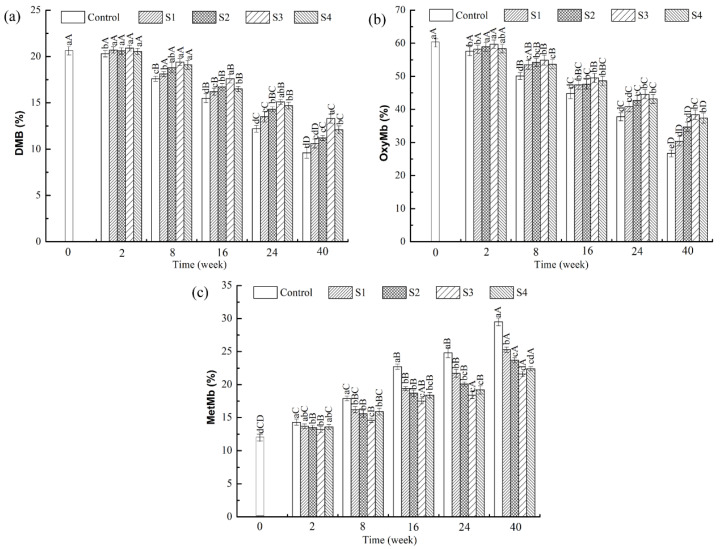
Contents of deoxymyoglobin (**a**), oxymyoglobin (**b**), and metmyoglobin (**c**) in lamb meat with different treatments during frozen storage. (MAP: microencapsulated apple polyphenols; Control: group without treatment; S1–S4: samples with 0.2, 0.8, 1.6, and 3.2 mg/mL of MAP). Letters (A–D) represent a significant difference (*p* < 0.05) between the frozen storage periods in the same treatments. Letters (a–d) represent a significant difference (*p* < 0.05) between the different treatments in the same frozen storage periods.

**Figure 3 foods-11-00537-f003:**
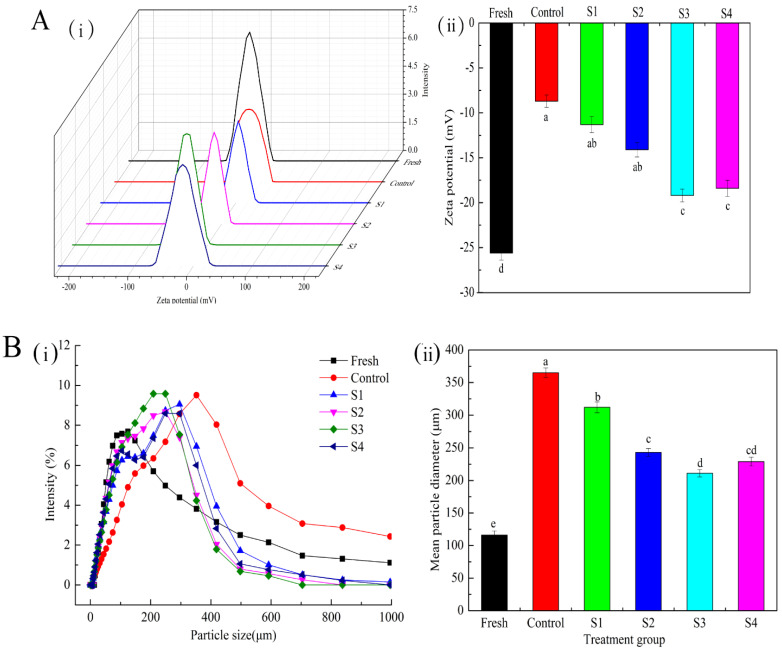
Zeta potential (**A**) and mean particle diameter (**B**) of MP affected by different treatments during frozen storage (40 weeks). A(i): Potential distribution; A(ii): Potential value; B(i): Particle size distribution; B(ii): Mean particle diameter. Means with different letters (a–e) within the same parameter group indicate a significant difference (*p* < 0.05). MAP: microencapsulated apple polyphenols; Fresh: Day 0; Control: group without treatment; S1–S4: samples with 0.2, 0.8, 1.6, and 3.2 mg/mL. Results are the average of triplicate measurements.

**Figure 4 foods-11-00537-f004:**
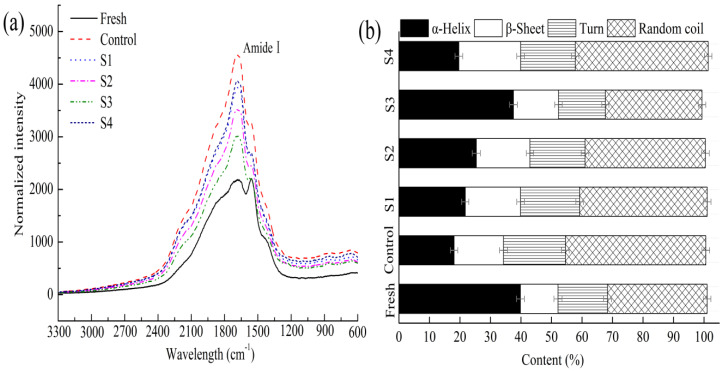
Secondary-structure fractions of MP with different treatments during frozen storage (40 weeks). (**a**): Raman spectra; (**b**): Content of protein secondary structure. MAP: microencapsulated apple polyphenols; Fresh: Day 0; Control: group without treatment; S1–S4: samples with 0.2, 0.8, 1.6, and 3.2 mg/mL of MAP. Results are the average of triplicate measurements.

**Figure 5 foods-11-00537-f005:**
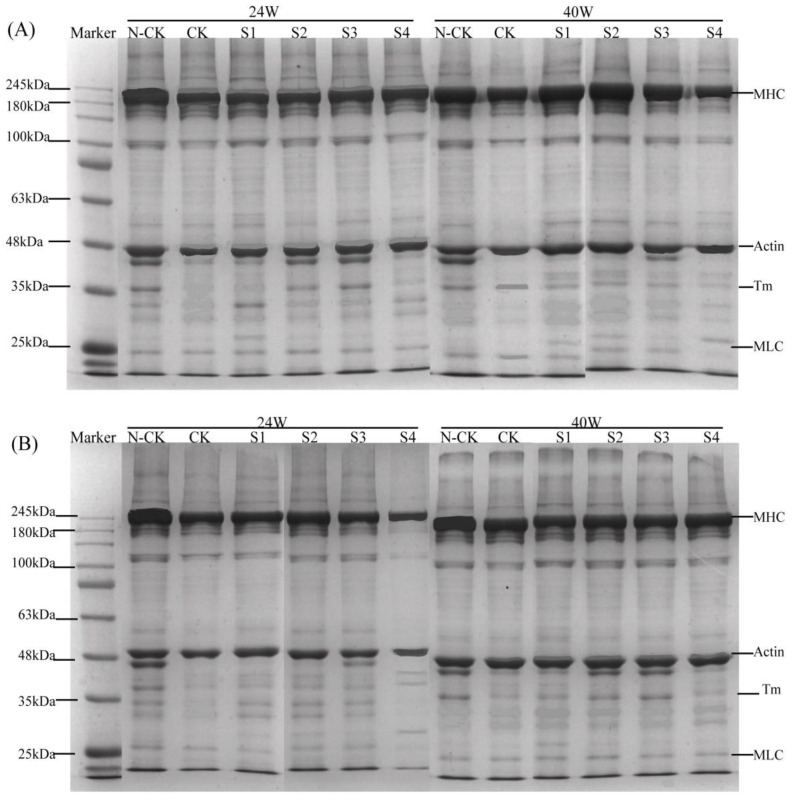
SDS–PAGE changes in myofibrillar proteins degraded by cathepsin in vitro during frozen storage ((**A**): Cathepsin B; (**B**): Cathepsin L; N-CK: Control group without cathepsin; CK: Control group added cathepsin; S1–S4: 0.2, 0.8, 1.6, and 3.2 mg/mL MAP-added samples with cathepsin; Detected bands; 1: myosin heavy chain (MHC; 210 kDa), 2: actin (Actin; 44 kDa), 3: tropomyosin (Tm; 35 kDa), 4: myosin light chain (MLC; 25 kDa)).

**Figure 6 foods-11-00537-f006:**
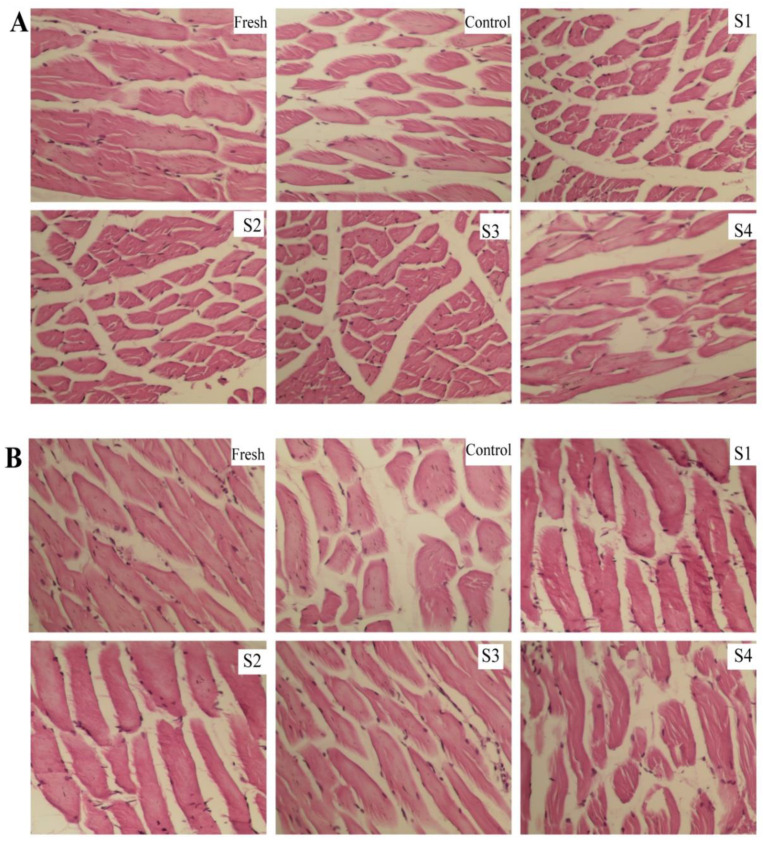
Results of hematoxylin–eosin staining of lamb after different treatments on changes in muscle tissue during frozen storage (Week 40). ((**A**): Cross-section; (**B**): Longitudinal section; MAP: microencapsulated apple polyphenols; Fresh: Day 0; Control: group without treatment; S1–S4: samples with 0.2, 0.8, 1.6, and 3.2 mg/mL of MAP).

**Table 1 foods-11-00537-t001:** Texture parameters measured during frozen storage of lamb meat with APM treatments.

	Weeks	Fresh	Control	S1	S2	S3	S4
Hardness (kg) Springiness (mm)	2 8 16 24 40 2 8 16 24 40	0.84 ± 0.02 ^aA^ 1.03 ± 0.01 ^aA^	0.81 ± 0.01 ^aA^ 0.76 ± 0.01 ^bB^ 0.61 ± 0.03 ^bcC^ 0.53 ± 0.02 ^dC^ 0.31 ± 0.04 ^dD^ 0.98 ± 0.02 ^aA^ 0.83 ± 0.03 ^cB^ 0.79 ± 0.01 ^cBC^ 0.74 ± 0.02 ^cC^ 0.63 ± 0.03 ^dD^	0.78 ± 0.02 ^bB^ 0.77 ± 0.03 ^bDB^ 0.7 ± 0.01 ^bB^ 0.65 ± 0.02 ^bcB^ 0.42 ± 0.03 ^cD^ 0.94 ± 0.01 ^bA^ 0.88 ± 0.02 ^bB^ 0.76 ± 0.02 ^cC^ 0.72 ± 0.01 ^cC^ 0.66 ± 0.02 ^cD^	0.81 ± 0.04 ^abA^ 0.76 ± 0.02 ^bCA^ 0.69 ± 0.01 ^bB^ 0.56 ± 0.03 ^cC^ 0.41 ± 0.04 ^cD^ 0.95 ± 0.02 ^bA^ 0.87 ± 0.01 ^bcB^ 0.79 ± 0.01 ^cC^ 0.76 ± 0.03 ^bcC^ 0.67 ± 0.02 ^cD^	0.86 ± 0.03 ^aA^ 0.79 ± 0.01 ^bA^ 0.73 ± 0.02 ^bA^ 0.66 ± 0.01 ^bcB^ 0.53 ± 0.02 ^bC^ 0.97 ± 0.02 ^aA^ 0.94 ± 0.01 ^aA^ 0.86 ± 0.02 ^bB^ 0.78 ± 0.02 ^bC^ 0.71 ± 0.03 ^bcC^	0.79 ± 0.02 ^bA^ 0.78 ± 0.04 ^bA^ 0.69 ± 0.03 ^bAB^ 0.62 ± 0.02 ^bcB^ 0.47 ± 0.02 ^bcC^ 0.95 ± 0.03 ^bA^ 0.88 ± 0.03 ^bB^ 0.82 ± 0.01 ^bcB^ 0.76 ± 0.02 ^bcC^ 0.68 ± 0.01 ^cD^

Notes: MAP: microencapsulated apple polyphenols; Fresh: Day 0; Control: group without treatment; S1–S4: samples with 0.2, 0.8, 1.6, and 3.2 mg/mL of MAP. Values with different letters (a–d) within a row of the same storage week indicate a significant difference (*p* < 0.05). Values with different letters (A–D) within a column of the same batch indicate a significant difference (*p* < 0.05). Results are the average of triplicate measurements.

## Data Availability

Data is contained within the article.

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
