# Peer review of "Improving the Quality of Frozen Lamb by Microencapsulated Apple Polyphenols: Effects on Cathepsin Activity, Texture, and Protein Oxidation Stability"

_foods, 2022, doi:10.3390/foods11040537_

Round 1

Reviewer 1 Report

The manuscript deals with improving the quality of frozen lamb by microencapsulated apple polyphenols: effects on cathepsin activity, texture, and protein oxidation stability.

The English language must be revised.

Please format all citations in the text according to the guide for authors.

Please number all equations.

Introduction

Line 67- “Several microencapsulated plant polyphenols have been investigated…”??Please specify.

Materials and methods

Line 97- “Group 1 received no treatment (control), whereas Groups 2–5 were soaked in MAP solutions of S1, S2, S3, and S4 for 60 min (4 °C) and then dried in a stainless-steel plate (5 min).”??only soaked??degree of penetration?only surface?use of vacuum impregnation??

Gain in solution with microcapsules??

TBARS analysis??

pH determination??

Color analysis??

Drip loss??

Mass loss??

SEM microcapsules??

Sensory analysis??

Results and discussion

Pictures of each sample?

Line 216- “To assess the effect of MAP treatment on the meat color of frozen lamb, changes in the contents of DMb, oxymyoglobin (OxyMb), and metmyoglobin (MetMb) were measured (Fig. 2).”??Figure 2, please add average values plus standard deviation.

Line 290- Figure 3, please identify each picture with different letters, rather than 3Bb.

Conclusion

Please do not repeat your results and focus on your main conclusions.

Reviewer 2 Report

References are not cited according to the authors instructions.

Materials and methods

Did you use market samples for the study?  How can you be sure about animal age and that conversion of muscle into meat was properly conducted?

Line 85: Please provide kit numbers.

Line 87: how did  you evenly distribute the samples among treatments and storage periods? Otherwise you cannot exclude variability within treatment and storage period.

Line 90: what do you mean with the term quality?  All you can see is marbling fat.

Line 90: How many samples?  Not several.

Please provide information on the type of packaging. Were the samples vacuum packaged?

Line 108: These authors evaluate meat colour with instrumental measurement. You are extracted the pigments.  Please correct.

Line 128: Extraction of myofibrillar proteins.  Please correct the method. As it is written it is not replicable.

Results and Discussion

In a large part of the text there is no information on the statistical differences between treatments (3.1 and 3.2).  There is an effort to explain the results but we do not know what is statistically different.

Please add p values in your graphs and tables.

Line 434: please be specific. Apple polyphenols not plant polyphenols that is general.

Round 2

Reviewer 1 Report

The manuscript was improved.

Author Response

  Response to Reviewer 1 Comments

        Thank you very much for your reviews, positive suggestions and recognition to our manuscript. Those comments are all valuable and very helpful for revising and improving our paper, as well as the important guiding significance to our researches.

Reviewer 2 Report

Meat pigments

How did you zero the spectrophotometer?

Line 188. All analyses were conducted in triplicate

3.1. Effect of different treatments on cathepsin activities in the frozen lamb

Please include in the discussion information on the results from the statistical analysis (p values)
